# Tackling of Immunorefractory Tumors by Targeting Alternative Immune Checkpoints

**DOI:** 10.3390/cancers15102774

**Published:** 2023-05-16

**Authors:** Dharmindra Dulal, Andrew Boring, David Terrero, Tiffany Johnson, Amit K. Tiwari, Dayanidhi Raman

**Affiliations:** 1Department of Cell and Cancer Biology, University of Toledo Health Science Campus, Toledo, OH 43614, USA; dharmindra.dulal@rockets.utoledo.edu (D.D.); andrew.boring@rockets.utoledo.edu (A.B.); amit.tiwari@utoledo.edu (A.K.T.); 2Department of Pharmacology & Experimental Therapeutics, College of Pharmacy & Pharmaceutical Sciences, University of Toledo Main Campus, Toledo, OH 43614, USA

**Keywords:** immuno-refractory tumors, CD8^+^ T lymphocytes, immunosuppression, immunotherapy, PD-1, immune checkpoints, CTLA-4, LAG-3, TIGIT, TIM-3, VISTA

## Abstract

**Simple Summary:**

Cancer treatment mostly involves drugs that has many side effects. In order to limit the toxicity of the chemical drugs, the scientific community embraced immunotherapy involving therapeutic antibodies targeting the tumor. In particular, immunotherapy aims to bolster the immune cells of the host (patients) to fight against cancer. Such therapies involve antibodies directed against traditional molecules in cancer and immune cells that prevent generation of autoimmunity or enable tolerance against self-antigens. This works in a subset of patients but develop resistance in others in the clinic. We suggest targeting other molecules on cancer and immune cells to overcome such resistance against immunotherapy. We discuss the possible resistance mechanisms against traditional immunotherapy and enlist new and alternative immunotherapy targets that can be employed as selective therapeutic interventions in the clinic along with low doses of chemotherapy. This will sustain and enhance the extraordinary momentum in cancer therapy.

**Abstract:**

Physiologically, well known or traditional immune checkpoints (ICs), such as CTLA-4 and PD-1, are in place to promote tolerance to self-antigens and prevent generation of autoimmunity. In cancer, the ICs are effectively engaged by the tumor cells or stromal ells from the tumor microenvironment through expression of cognate ligands for the ICs present on the cell surface of CD8^+^ T lymphocytes. The ligation of ICs on CD8^+^ T lymphocytes triggers inhibitory signaling pathways, leading to quiescence or an exhaustion of CD8^+^ T lymphocytes. This results in failure of immunotherapy. To overcome this, several FDA-approved therapeutic antibodies are available, but the clinical outcome is quite variable due to the resistance encountered through upregulated expression of alternate ICs such as VISTA, LAG-3, TIGIT and TIM-3. This review focuses on the roles played by the traditional as well as alternate ICs and the contribution of associated signaling pathways in generating such resistance to immunotherapy. Combinatorial targeting of traditional and alternate ICs might be beneficial for immune-refractory tumors.

## 1. Introduction

Breast cancer (BC) has a high morbidity rate and was declared as the most common cause of death among the cancer types in the United States in 2020. In 2021 alone, the incidence of cancer in the US was approaching 2 million with upwards of 600,000 deaths [1]. It has been estimated that in the US, 30% of all new cases of cancer in women will be of BC [2]. Moreover, more than 12% of women will be diagnosed with BC over their lifetime [2]. Mortality is significantly higher when breast cancer metastasizes to other organs such as the lungs, bones and brain, especially in triple-negative breast cancer (TNBC) [3]. Metastasis is responsible for most cancer-related deaths; however, the intricate mechanism is poorly understood, leading to therapy failure.

The interplay between cancer and the immune system has proven to be significant in primary cancer progression and metastasis. The tumor microenvironment (TME) is strongly associated with driving tumorigenicity and disease progression [4,5]. The TME is composed of stromal cells including the adaptive and innate immune cells, fibroblasts, adipocytes, mesenchymal cells, endothelial cells, lymphendothelial cells, and the acellular extracellular matrix [6,7]. Stromal mesenchymal cells have been implicated in secreting immunosuppressive molecules which can promote cancer development, as well as control the innate and adaptive immune response [8]. Both the adaptive and innate immune cells contribute to the recognition of foreign pathogens and tumors. Adaptive immune cells are broadly categorized into T and B lymphocytes. T and B lymphocytes have diversified receptors for recognizing antigen which contribute to specificity, recognition, and response to various antigenic epitopes [9,10]. T lymphocytes can be distinguished into two different categories. Cytotoxic, CD8^+^ T lymphocytes can directly target cancer cells for destruction by recognizing abnormal appearing antigens (neoantigens) on their surface [6]. In addition to direct targeting for destruction, cytotoxic CD8^+^ T lymphocytes also secrete soluble cytokines. Concurrent secretion of interferon gamma (IFN-*γ*) and tumor necrosis factor (TNF) by cytotoxic CD8^+^ T lymphocytes leads to a compromise in the integrity of tumor-resident blood vessels, and an ischemic state of the tumor is generated and sustained [11,12]. CD4^+^ T lymphocytes can be divided into subcategories of helper T lymphocytes, including Th1, Th2, and Th17. Through the production of cytokines, these helper CD4^+^ T lymphocytes can function in maintaining and enhancing the activity of the cytotoxic CD8^+^ T lymphocytes; thus, they exhibit anti-tumor immune responses [13,14]. Additionally there are cytotoxic, CD4^+^ T lymphocytes, and regulatory T cell subsets (T_regs_) [15]. While cytotoxic CD8^+^ and CD4^+^ T lymphocytes directly contribute to tumor cytotoxicity through secretions of perforin and granzyme cytokines, T_regs_ have the opposite effect [14]. T_regs_ have been shown to ultimately promote tumor growth and development by decreasing the effector function of T lymphocytes that infiltrated a tumor [6,14]. These adaptive immune cells are effective; however, they are not the only tumor defense mechanism. Innate immune cells are also associated with cancer. While they lack the diversified antigen receptors that adaptive immune cells have, many innate lymphoid cells produce cytokines. Innate immune cells can respond to commensals and pathogens at mucosal barriers, help potentiate adaptive immunity, and participate in the regulation of tissue inflammation [16]. Tumor cells can evade both adaptive and innate immune anti-tumor responses through various mechanisms. This leads to adaptive immune resistance, subsequently promoting cancer growth [17]. One of the ways in which adaptive immune resistance occurs is through metabolic nutrient alterations in the TME. The TME requires a vast amount of metabolic nutrients, the contents of which can determine the activation or suppression of cellular immunity. Depletion of extracellular glucose due to increased consumption by tumor cells is shown to restrict glucose availability to T lymphocytes, thus decreasing their effector functions [18]. When the TME is exhausted of its necessary nutrients, it can become immunosuppressive and lead to the progression of cancer [19]. Additionally, there are several immune checkpoints (ICs) that allow for inhibition of activating the immune system. These checkpoints have been shown to limit the anti-tumor response of the immune system. Inhibition of these immune checkpoints have become effective targets in several malignancies [20,21].

Alterations in genetic elements or modulation of the TME dictate primary tumor progression and metastasis. Over time, these alterations result in the deactivation of regulatory molecules, namely IC proteins. IC proteins are vital regulators of immune response against foreign antigens and simultaneously maintain tolerance against self-antigens. The classical IC consists of co-inhibitory cell surface receptors on CD8^+^ cytotoxic T-lymphocytes such as cytotoxic T lymphocyte associated protein 4 (CTLA-4) and programmed cell death protein 1 (PD-1). Blocking CTLA-4 and PD-1 with therapeutic antibodies was found to provide survival benefit to cancer patients by rejuvenating the anti-tumor response of CD8^+^ cytotoxic T-lymphocytes.

New IC proteins have been discovered, including the V-domain Ig suppressor of T cell activation (VISTA), T cell immunoreceptor with Ig and immunoreceptor tyrosine-based inhibitory motif (TIGIT), lymphocyte-activation gene 3 (LAG-3), T cell immunoglobulin and mucin-domain containing 3 (TIM3) and others. Tumors not only disrupt the functional surveillance by ICs to avoid initiation of apoptotic clearance but also contribute to therapeutic resistance through the upregulation of other ICs. This process protects tumor cells by warding off the anti-tumor response from the immune system especially from CD8^+^ cytotoxic T lymphocytes [22]. These interactions between different ICs are depicted in Figure 1.

VISTA receptor is a relatively new IC discovered in 2011 [23]. After its identification, the role of VISTA in tumor progression and metastasis has been studied intensively. VISTA has two ligands and its affinity for these ligands depends on the acidity of the cellular environment [24]. The TME is usually acidic, and this could center VISTA as an actionable molecular target. Currently, several clinical trials are being conducted to determine the efficacy of anti-VISTA therapeutic antibodies. TIGIT was discovered in 2009 by three different laboratories [25,26,27] and the operative mechanism of TIGIT is not fully understood yet. TIGIT has been implicated in tumorigenesis and metastasis however, and it is being evaluated clinically as an effective therapeutic target. LAG-3 was discovered in 1990 in natural killer (NK) cells [28]. The discovery of its ligand, major histocompatibility complex II (MHCII) has further enhanced the understanding of how LAG-3 downregulates the functioning of CD4^+^ T lymphocytes [29]. There are several ongoing clinical trials targeting LAG-3 in human tumors. TIM-3 was discovered in 2002 on the cell surface of Th1 T lymphocytes [30] and was described as a negative regulator of immune cells. As TIM3 levels were downregulated when Th1 T lymphocytes were activated [30]. The most prominent IC is PD-1. PD-1 was originally discovered in 1992 on the plasma membrane of CD8^+^ T lymphocytes [31]. Although it was thought to be involved in programmed cell death, the mode of action was unknown until the discovery of its ligand, program cell death ligand 1 (PD-L1) in 1999 [31,32]. Upon observation that PD-L1 suppresses the functionality of CD8^+^ T lymphocytes following ligation to its receptor, PD-1, it was concluded that PD-1 is an IC [33]. Moreover, the elevated levels of PD-L1 on tumor cells suggested that PD-L1/PD-1 axis would be excellent therapeutic targets for oncological drugs [34]. There are a variety of signaling pathways implicated in the induction of ICs and this is depicted in Figure 2.

In addition to the aforementioned ICs, several ICs have also been implicated in cancers, such as CTLA-4 and B and T lymphocyte attenuator (BTLA). In this review, the role of five ICs, viz., VISTA, TIGIT, LAG-3, TIM-3, and PD-1, will be examined in cancer progression, metastasis, and clinically refractory cases. The roles of these ICs, as well as recent developments in treatments targeting these ICs, and different mechanisms of therapeutic resistance are discussed. Although the discussion applies to many human tumors, our primary emphasis in this review is breast cancer, especially TNBC.

## 2. VISTA

As we lean more towards immunotherapy as opposed to chemotherapy for the treatment of oncological diseases, the targetable ICs such as V-domain Ig suppression of T Cell activation (VISTA) has garnered much attention [25,26,27]. VISTA is a type 1 transmembrane protein with a N-terminal Ig V domain of 30 amino acids, a transmembrane domain, and a cytoplasmic domain of 95 amino acids [23]. The VISTA gene is located on the long arm of chromosome 10 (10q22.1), away from the other Ig superfamilies. It can bind to different ligands based on the acidity of the cellular microenvironment, which differentiates it from the traditional B7 family of receptors [24,35].

The density of the expression of VISTA is elevated in the cells of myeloid lineages, such as macrophages, dendritic cells, monocytes in tissues, and circulating monocytes [36]. Although VISTA is highly expressed on mature antigen-presenting cells (APCs), its role in the quiescence of naïve T lymphocytes, especially naïve CD4^+^ T lymphocytes, is critical in maintaining peripheral tolerance to self-antigens [36]. The role of VISTA was further strengthened by the study conducted on VISTA-deficient mice which resulted in increased proliferation of memory CD4^+^ T lymphocytes and related cytokines causing autoimmunity and inflammatory disease [36]. VISTA regulates quiescence of T lymphocytes by upregulating anti-proliferative genes such as Klf2, Klf6, Ccr7, Btg1, and Btg2 [37,38,39]. Moreover, VISTA is also known for maintaining epigenetic programs for regulation of naïve T lymphocytes. Therefore, the loss of VISTA has been linked to an increase in accessibility to promoters of proliferative genes, such as Tcf7, Ifngr1, Bcl2, and Il7ra, causing a breakdown in T lymphocyte peripheral tolerance [36]. Another paradigm of VISTA is its capacity to function as both a ligand and a receptor. It has been proposed that VISTA acts as a ligand when expressed on APCs, thereby engaging an inhibitory receptor on T lymphocytes, and suppressing their proliferation [40]. On the contrary, VISTA expresses itself as both a ligand and a receptor on the surface of T lymphocytes, resulting in homotypic interaction/signaling or engaging with other receptors, such as V-Set and Immunoglobulin domain containing 3 (VSIG3) and P-selectin glycoprotein ligand-1 (PSGL1) [40,41].

As mentioned earlier, while reduced expression of VISTA results in autoimmune and inflammatory diseases, overexpression can lead to the development and progression of cancers by limiting T lymphocyte proliferation. One mechanism by which VISTA promotes cancer progression is via its receptor VSIG3, which is found endogenously but is elevated in cancer cells [35]. VSIG3 is a type 1 transmembrane receptor mostly expressed in the brain and testis-specific IgG family [42]. In normal physiology, VSIG3 functions to mediate homophilic adhesion of cells in a calcium-independent manner and protects the integrity of the blood-testis barrier [35,42]. Although the mechanism governing the increased expression of VSIG3 on cancer cells is still unknown, its interaction with VISTA to reduce the proliferation of CD4^+^ T lymphocytes is evident [35]. Upon the interaction with VSIG3, VISTA sends downstream signals in the T lymphocytes inhibiting the production of IFN-γ, IL-2, IL-17, CCL5, CCL3, and CXCL11 [35]. These cytokines play vital roles in the immune response in tumors by increasing T lymphocyte proliferation and infiltration of immune cells into the TME [35]. Therefore, inhibition of the VISTA-VSIG3 axis will lead to an increase in the levels of these beneficial cytokines and chemokines. This promotes more immunosurveillance within the TME by limiting tumor growth and increasing tumor clearance.

The environment of the tumor bed is more acidic than the physiological pH. Under these acidic conditions, VISTA acts as a selective ligand for PSGL1 [24]. PSGL1 is expressed on the surface of most hematopoietic cells and is responsible for the migration of leukocytes to the inflammatory sites via binding with P-selectin on vascular endothelium [43]. Although post-translational modifications such as glycosylation and sulfation, are required for PSGL1 binding to P-selectin, it is absent in naïve T lymphocytes. PSGL1 can engage with other ligands, such as VISTA, functioning as a negative regulator for T lymphocyte responses [44]. This interaction between VISTA and PSGL1 is significantly pronounced in an acidic environment where histidine residues H152, H154, and H155 are deprotonated [24]. It has been hypothesized that interaction between VISTA and PSGL1 in the TME inhibits the proliferation of T lymphocytes and causes exhaustion of T lymphocytes. However, the exact mechanism behind these processes has yet to be determined [24].

Recent analysis of the TME has revealed a significant increase, up to ten-fold, in expression of VISTA in myeloid-derived suppressor cells (MDSCs) [45]. MDSCs are pathologically activated, immature myeloid cells of either macrophage or neutrophil lineage. MDSCs have potent immunosuppressive activities and are pro-tumorigenic in nature. The signaling mechanism of how VISTA inhibits the proliferation of CD4^+^ T lymphocytes in the TME is yet to be determined. Blocking VISTA receptor-ligand interactions has been shown to increase the migration of tumor-specific effector T lymphocytes into the TME [45]. This presumably occurs via increasing concentrations of IFN-γ and CCL5 in the TME [45]. Moreover, the enhanced expression of MHC II and CD80, along with the production of anti-tumor cytokines, such as IL-12 and TNF-α, was observed in the TME (14). Blockade of VISTA also results in impairment of the immunosuppressive effect of T_regs_ [45]. Recent analysis of a cohort of 919 BC patients has revealed the VISTA positivity in immune cells and tumor cells in 29.1% and 8.2% of patients, respectively [46]. Moreover, the increase in VISTA in TNBC was correlated with unfavorable clinicopathological outcome in TNBC patients [46]. Among immune cells, VISTA was found to be expressed highest in CD68^+^ macrophages, 32%, in BC [47]. Suggesting the possibility of targeting VISTA as part of a therapeutic effort. Therefore, many clinical trials have been initiated to determine the clinical efficacy of VISTA blockade. Some of these clinical trials are tabulated in Table 1.

## 3. TIGIT

The role of IC proteins in cancer can be the difference between prevention or progression of a cancer and has opened a new avenue of potential therapeutic options for many patients. One of the ICs that plays a vital role in this process is TIGIT, also known as Washington University cell adhesion molecule (WUCAM), V-set and transmembrane domain-containing protein 3 (Vstm3), and V-set and immunoglobulin domain-containing protein 9 (VSIG9). TIGIT is a transmembrane receptor that belongs to the PVR-like protein family and is comprised of 244 amino acid residues whose gene is located on the q-arm of the chromosome 3 [51]. Its structure consists of one immunoglobulin variable (IgV) domain, a type 1 transmembrane domain, and a cytoplasmic tail [25,26,27]. The cytoplasmic tail contains an immunoreceptor tyrosine-based inhibitory motif (ITIM) and the immunoglobulin tyrosine tail (ITT)-like motif [25,26,27,52]. In normal physiological conditions, TIGIT is expressed as a receptor on most NK cells and T lymphocytes, including helper and cytotoxic T lymphocytes, and T_regs_ [25,26,27].

Although TIGIT can bind to CD155 (also called poliovirus receptor; PVR), CD112 (also called poliovirus receptor-related 2; PVRL2), and CD113, it binds with CD155 with the highest affinity, making it the primary ligand for TIGIT [26,53]. All of these ligands belong to the family of nectin and nectin-like (NECL) proteins. Nectin and NECL are responsible for mediating cell adhesion, signaling, and polarization [54]. CD112 is an adhesive protein that can act as either a stimulator or an inhibitor depending on the receptor [53]. For example, the interaction of CD122 with DNAX accessory molecule-1 (DNAM-1 or CD226) stimulates T lymphocyte proliferation and cytokine production, whereas the binding of CD122 with TIGIT dampens T lymphocyte proliferation and their response to stimuli [55,56]. Normally CD112 is found in the bone marrow, kidneys, pancreas, and lung cells [53]. Another ligand of TIGIT, CD113, is responsible for establishing and maintaining cell-cell junctions along with its role in cell adhesion and signaling [57]. As with CD112, overexpression of CD113 also leads to exhaustion of T lymphocytes [57]. The primary ligand of TIGIT, CD155, has a specific “lock-and-key” motif designated for TIGIT binding [58]. CD155 is expressed on macrophages, human dendritic cells, B lymphocytes, epithelial cells, and tumor cells [58]. Upon activation, TIGIT binds to CD155 in a cis-trans configuration [58]. Then two TIGIT-CD155 dimers join to form a heterotetramer [58]. In this heterotetramer, there is TIGIT-TIGIT binding in cis configuration and each TIGIT protein in this cis state binds to CD155 in *trans* configuration [58]. All of these ligands are highly expressed in tumors of the breast, bladder, lung, and pancreas [59,60,61]. While under normal physiological conditions, the interaction of TIGIT with these ligands prevents the self-destruction of normal cells by NK cells; however, in the TME, these ligands bind to TIGIT in T lymphocytes and suppress immunological responses by altering the signaling cascade in T lymphocytes [62].

TIGIT can mediate the innate and adaptive immune system via impacting the activities of T lymphocytes and NK cells. In the cell-extrinsic mechanism, TIGIT binds to the CD155 ligand on the surface of dendritic cells (DCs). This interaction impedes the ability of DCs to process and present the antigen, thus failure to propagate appropriate T lymphocyte response [27]. Furthermore, upon activation of the TIGIT-CD155 axis, the secretion of the anti-inflammatory cytokine, IL-10, from DCs is favored over the secretion of pro-inflammatory cytokines, IL-12, further limiting the activation of T lymphocytes [27]. In the cell-intrinsic mechanism, TIGIT works via its cytoplasmic tail. The binding of the ligand, CD155, in the extracellular domain leads to phosphorylation of Tyr231 and Tyr 225 in the ITIM and ITT motif of the cytoplasmic tail, respectively [62]. This phosphorylation promotes the recruitment of two cytosolic adaptor proteins, growth factor receptor-bound protein 2 (Grb2) and β-arrestin2 by the cytoplasmic tail [62,63]. Grb2 then recruits *Src* homology 2 (SH2)-containing inositol phosphate-1 (SHIP1), whose function is to inhibit phosphoinositide-3-kinase (PI3K) and mitogen-activated protein kinase (MAPK) signaling cascades [63]. Furthermore, the recruitment of β-arrestin2 leads to suppression of autoubiquitination of TNF receptor-associated factor 6 (TRAF-6), inhibiting nuclear factor kappa B (NF-κB) activation [62,63]. Inhibiting PI3K/MAPK signaling results in downregulation of NK cell killing activity, whereas inhibition of NF-κB impairs the production and secretion of IFN-γ [62].

Another mechanism by which TIGIT is implicated in tumor progression is via its competitive and direct inhibition of DNAM-1. DNAM-1 is a cell surface adhesive glycoprotein, mostly expressed by NK and T lymphocytes, and a receptor for ligand CD155 and CD112 [64,65]. During infection or in the TME, DNAM facilitates the adhesion of immune cells by interacting with CD155 and CD112 [64,65]. Furthermore, the interaction of DNAM-1-CD155/CD112 promotes the release of cytotoxic molecules from NK Cells, thereby eliminating the infected/tumor cells [64,65]. This normal process is disrupted when overexpression of TIGIT outcompetes DNAM-1 for its binding to CD155, limiting NK and T lymphocytes cytotoxic activities against tumor cells [27]. Moreover, TIGIT can directly interact with DNAM-1 on the cell surface, impairing its ability to bind to its designated ligand [27]. In addition, TIGIT is also known for its role in modulating T_reg_ response [66]. The overexpression of TIGIT in T_reg_ is correlated with higher expression of suppressive genes, such as Foxp3, CD23, and CTLA-3 [66]. This expression signature led to suppression of Th1 and Th17 cell responses [66].

In the TME, there is an overexpression of CD155 by tumor cells [67,68,69,70]. As NK and T lymphocytes enter the TME, they express TIGIT on their surface that interacts with the CD155 on tumor cells [67,68,69,70]. CD155 on tumor cells normally interacts with CD266 expressed by NK and T lymphocytes, but this binding is disrupted due to the higher affinity of CD155 for TIGIT [67,68,69,70]. This interaction between TIGIT and CD155 will result in deactivation of NK and T lymphocytes [67]. Moreover, the interaction of CD155 on tumor cells and TIGIT on T_reg_ further hampers the normal immune response. Several studies have suggested the correlation between increased expression of CD155 on tumor cells and aggressiveness of BC [68,69,70]. TIGIT has been found to be significantly elevated in invasive BC compared to normal tissue [68]. In a mRNA analysis of a cohort of 197 BC patients, an elevated level of CD115 was significantly correlated with poor prognosis and decreased recurrence-free survival rates [69]. Moreover, the blockade of TIGIT on immune cells or CD155 on cancer cells was shown to promote the cytotoxic effects of immune cells [69]. The vital influence of TIGIT on the TME has encouraged the development of therapeutic drugs. There are several clinical trials in progress targeting TIGIT and its receptor PVR to halt tumor progression. Some of these clinical trials targeting BCs are depicted in Table 2.

## 4. LAG 3

Tumor cells can also evade immune surveillance through manipulation of the IC, LAG-3, which acts as a negative regulator of tumor infiltrating lymphocytes (TILs). LAG-3 is encoded by the *lag-3* gene located on the distal part of the short arm of chromosome 12 [29]. This gene encodes a type I membrane protein of 498 amino acid residues (50 kDa) [29]. The locus of the *lag-3* gene is adjacent to the CD4 gene, signifying the relationship in the evolutionary origin and similarities in their functions [28]. Structurally, LAG-3 consists of three regions, extracellular, transmembrane, and intracellular domains [28]. The extracellular region of LAG-3 has four immunoglobulin superfamily-like (IgSF) domains (D1–D4) [82], the most important of which is the D1 domain because of its Ig variable-like region (V-SET type) which contains a proline-rich loop and intrachain disulfide bridge responsible for LAG-3-MHC II interaction [28,82]. The intracellular domain of LAG-3 has three conserved motifs, a serine residue (S484) which can be phosphorylated and regulated, a KIEELE motif, and a glutamate-proline dipeptide multiple repeats motif (EP motif) [83]. The phosphorylation of S484 has been attributed to the downstream signaling leading to the production of the cytokine IL-2 [83]. The KIEELE motif is a highly conserved motif found in LAG-3 [84,85]. The implication of lysine (K468) residue of KIEELE motif is found to be vital in LAG-3 inhibitory downstream signaling [84,85]. The EP repeat motif in the intracellular portion of LAG-3 is important in LAG-3 colocalization with CD3, CD4 and CD8 proteins and activation of the T-cell receptor (TCR) within lipid rafts [86]. However, the deletion of EP motif had no effect on the downstream signaling of LAG-3 [84].

LAG-3 is expressed on the plasma membrane of the activated CD4^+^ T helper lymphocytes, except for Th2, cytotoxic CD8^+^ T lymphocytes, and T_reg_, NK cells, B lymphocytes, and DCs [87]. The expression of LAG-3 is elevated by cytokines such as IL-2 and IL-10 [88,89]. The activation of LAG-3 results in increased production of IL-10 by T lymphocytes [90]. There are multiple ligands for LAG-3 including MHC class II, galectin-3 (Gal-3), and fibrinogen-like protein 1 (FGL1). MHC II interacts with LAG-3 with higher affinity than CD4 via its D1 domain [29]. Once bound to MHC II, the cytoplasmic domains of LAG-3 propagate inhibitory signals in CD4^+^ T lymphocytes, leading to its inactivation and the escape of tumor cells from undergoing immune cell-mediated apoptosis [91]. Another ligand of LAG-3 is Gal-3, a 31-kDa galactose-binding lectin that is responsible for broad range of physiological functions such as cell differentiation, apoptosis, cell adhesion, chemoattraction, and cell cycle progression [92]. Gal-3 is found in different cellular compartments including the cytoplasm, nucleus, and on the extracellular surface [92]. When expressed on the surface of tumor cells in the TME, Gal-3 interacts with LAG-3 on CD8^+^ T lymphocytes inhibit cytotoxicity of CD8^+^ T lymphocytes [93]. FGL1 is a protein produced by the liver which functions to maintain a tolerogenic environment in the liver [94]. Recently, FGL1 was determined to be significantly upregulated in solid tumors, including TNBC [94]. The fibrinogen-like domain of FGL1 interact with D1 and D2 domains of LAG-3, producing immunosuppressive downstream signaling in T lymphocytes [94].

In normal physiology, LAG-3 plays a critical role in regulating autoimmunity as its function is to limit overactivation of T lymphocytes. The expression of LAG-3 is also tightly controlled by various membrane-bound metalloproteases ADAM10 and ADAM17 [95]. To mount the immune response, ADAM10 cleaves the connecting peptide of LAG-3 to facilitate T cell activation [95]. The activity of ADAM10 was found to be 12-fold higher following T lymphocyte activation. ADAM17 mediates LAG-3 cleavage in a separate protein kinase C-dependent manner following TCR signaling [95]. In the TME, the downregulation of these metalloproteases and upregulation of cytokines such as IL-12 and IL-2 increases the expression of LAG-3, which results in the unregulated inhibition of T lymphocytes. The study of LAG-3 in BC samples of a cohort of 2994 patients revealed a significant elevation of LAG-3 in the TME of BC compared to normal breast tissue [96]. Moreover, the stage of cancer is positively correlated with the concentration of LAG-3 present in the TME [96]. Therefore, it was concluded that LAG-3 could be an important prognostic marker for TNBC [96]. Along with being a prognostic marker, the role of LAG-3 in the tumor metastasis makes it a good therapeutic target for the treatment of cancer. Therefore, multiple drugs are in clinical trials targeting the LAG-3 and its ligands and is provided in Table 3.

## 5. TIM-3

One emerging IC that it is important to acknowledge is the role of TIM-3 in the TME. TIM-3 is a transmembrane protein that is part of TIM family of immunoregulatory proteins [30]. TIM-3 is encoded by the *tim-3* (*HAVCR2*) gene located on the long arm of chromosome 5 [30]. The structure of TIM-3 consists of an amino-terminal immunoglobulin variable domain (V domain), a mucin stalk, a transmembrane domain, and a cytoplasmic tail [30]. Initially, TIM-3 was thought to be expressed only on T lymphocytes. However, numerous studies have suggested that TIM-3 is also expressed not only on T lymphocytes but on a variety of immune cells, such as macrophages, mast cells, NK cells, and tumor-associated DCs [30,99,100,101]. The function of TIM-3 varies depending on its expression in different cell types. For example, in macrophages, TIM-3 is responsible for regulating the response to toll-like receptor 2 (TLR2) and toll-like receptor 4 (TLR4) stimulation, thus inhibiting the release of proinflammatory cytokines in autoimmune diseases [102,103]. In mast cells, TIM-3 plays a role in cellular activation upon IgE sensitization and antigen presentation [104]. Dysfunction of TIM-3 on the surface of NK cells in the TME often leads to escape of tumor cells from the anti-tumor immune response [101].

TIM-3 has been reported to have additional ligands such as galectin 9 (Gal-9), phosphatidylserine (PtdSer), carcinoembryonic antigen-related cell adhesion molecule 1 (CEACAM1), and high mobility group protein B1 (HMGB1). Gal-9 is a C-type lectin that is secreted by multiple hematopoietic cells, tumor cells, APC cells, and TIM-3 expressing cells in an autocrine fashion. It is expressed on the surface of tumor cells as well [105]. The binding of Gal-9 on the carbohydrate motif of the V-domain promotes the oligomerization of TIM-3, thus facilitating the binding of another TIM-3 ligand, such as CEACAM1 [106,107]. Initially, when TIM-3 is not bound to Gal-9, amino acid residues Tyr256 and Tyr263, in the cytoplasmic domain interact with HLA-B-associated transcript (BAT3) protein [106,107]. BAT3 is a negative regulator of TIM-3, inhibiting the inhibitory function of TIM-3. However, as Gal-9 interacts with TIM-3, it triggers the phosphorylation of Tyr256 and Tyr263 which releases the BAT3 from the cytoplasmic domain of TIM-3 [108]. This results in unregulated inhibition of T lymphocytes by TIM-3. Another ligand of TIM-3 is PtdSer, an apoptotic cell surface marker. PtdSer binds the pocket of the V-domain on TIM-3 formed by the FG and CC loops [109]. This interaction between TIM-3 and PtdSer is not implicated in apoptosis or T lymphocyte anergy. However, it is found to be crucial in antigen presentation by TIM-3 expressing DCs [110]. The mechanism of action of CEACAM1 upon interaction with TIM-3 is similar to that of Gal-9. CEACAM1 is expressed by T lymphocytes, macrophages, DCs, monocytes and tumor cells [111,112]. The binding of CEACAM1 on the FG and CC loop of TIM-3 promotes the release of BAT3 via phosphorylation of the Tyr256 and Tyr263 residues [108]. Moreover, recent studies have suggested that CEACAM1 can also bind TIM-3 in intracellular compartments [106]. The complex mechanism of TIM-3-CEACAM1 interaction in the TME hasn’t been fully understood. The HMGB1 is an intranuclear protein responsible for binding to DNA helix and facilitating with the formation of protein complexes [113]. During cell stress or death, HMGB1 is released by the cells thus functioning as a marker for cell death, stress, or inflammation [113]. HMGB1 is expressed by tumor cells in the TME. The binding of HMGB1 with TIM-3 on T lymphocytes and macrophages induces the release of proinflammatory cytokines such as IL-1, IL-6, and IL-8, creating an inflammatory TME [114]. These inflammatory conditions in turn facilitate angiogenesis and enable further tumor progression such as metastasis.

In normal physiological conditions, TIM-3 is vital in the regulation of the immune system and prevention of autoimmune diseases. For example, blockade of TIM-3 via anti-TIM-3 antibodies resulted in the hyperactivation of immune cells, especially macrophages and Th1 cells [115]. TIM-3-deficient mice were found to develop autoimmune diseases more often than normal mice [116]. Additionally, the expression of TIM-3 was downregulated in patients with multiple sclerosis, rheumatoid arthritis, psoriasis, and ulcerative colitis [116,117,118,119]. TIM-3 on T_reg_ was determined to play an influential role in suppression of the allograft rejection [120]. While downregulation of TIM-3 causes autoimmune diseases, upregulation is implicated in T lymphocyte exhaustion. This leads to cancer progression and metastasis. Recent studies performed on BC tissue have implicated the expression of TIM-3 on cancer progression and metastasis. The expression of TIM-3 on tumor cells or TILs is directly correlated to the staging of BC sample [121,122]. Furthermore, TIM-3 was found to directly correlate with Ki67 (a cell proliferation marker) and 5-year patient survival rate (poor prognosis) [121,122]. The prominent role for TIM-3 in tumor progression has made it an attractive therapeutic target. Therefore, multiple clinical studies are ongoing to determine the potential of TIM-3 as an immune checkpoint inhibitor (ICI) and they are shown in Table 4.

## 6. PD-1

Programmed Cell Death Protein 1 (PD-1) is a co-inhibitory receptor expressed on the surface of T lymphocytes as part of the TCR complex with two B7 like ligands [132]. PD-1 and its two ligands programmed cell death ligand 1 and 2 (PD-L1/2) form one of the most well documented and targeted ICs in cancer biology. Initially discovered in mice, human PD-1 was found to be encoded by the *pdcd1* gene and maps to chromosome 2 (2q.37.3) by in situ hybridization. PD-1 has a molecular mass of 55 kDa and is comprised of 288 amino acid residues. Structurally, PD-1 contains an extracellular IgV-like N-terminal domain [133,134]. PD-1 shares a sequence similarity of 20% to another IC; CTLA-4, the first IC that was approved for targeted therapy by the FDA in 2011. However, targeting CTLA-4 with the monoclonal antibody Ipilimumab showed promising activity against melanoma, but patients receiving this treatment experienced adverse immune-mediated events [135]. These results demonstrated both the power of therapies targeting ICs and the need for additional immunotherapy targets.

PD-1 is expressed on the surface of a wide range of activated immune cells including dendritic cells, B lymphocytes, macrophages, monocytes, NK cells, and most importantly tumor-specific T lymphocytes [136]. In healthy individuals, the expression of PD-1 and its binding with the PD-L1 ligand determines the extent of T lymphocyte activation and balances autoimmunity with self-tolerance [33]. Binding of PD-1 with PD-L1 prevents TCR mediated proliferation of T lymphocytes and impairs effector cytokine production [33,136]. The inverse of this concept has been further demonstrated as mice lacking the PDCD1 gene were susceptible to autoimmune diseases due to proliferation of self-reactive T lymphocytes [137]. Expression of PD-1 is under control of many different transcription factors such as nuclear factor of activated T lymphocytes (NFAT), NOTCH, Forkhead box protein 01 (FOX01) and interferon regulatory factor 9 (IRF9) [138]. Transcription of PD-1 is also under epigenetic control, and in chronic infections, T lymphocytes become functionally exhausted, and express high levels of PD-1 [139].

Factors within the TME also regulate this IC. Tumor associated macrophages (TAMs) have been classified into anti-tumor M1 or pro-tumor M2 categories. M2 macrophages secreting pro-inflammatory cytokines have been shown to increase PD-1/PD-L1 interactions and expression [140,141]. In several types of cancer, increased infiltration of M2 macrophages into the TME correlates with a poor prognosis [142]. Through these mechanisms, TAMs have been thought to contribute to resistance to anti-PD-1/PD-L1 therapies and prevent T lymphocytes from mounting an anti-tumor response [143,144]. By depleting M2 macrophages, some of this resistance can be overcome [145].

Binding of PD-1 with PD-L1 prevents the immune system from mounting an immune response, even when the TCR recognizes an actionable antigen [146]. Many cancers employ the PD-1/PD-L1 axis to avoid immune surveillance by expressing PD-L1 on the tumor cell surface, although the value of PD-L1 expression as a prognostic indicator is variable. One review found PD-L1 expression was correlated with a poor prognosis in bladder cancer, esophageal cancer, gastric cancer, hepatocellular cancer, renal cell carcinoma, ovarian cancer, and pancreatic cancer, and a good prognosis Merkel cell carcinoma [147]. Even if the immune axis expression is not a good predictor for prognosis, inhibitors of the PD-1/PD-L1 axis are indicated in various cancer types. Three anti-PD-1 antibodies, nivolumab, pembrolizumab, and cemiplimab, have been approved by FDA for a wide range of cancers, along with three anti-PD-L1 monoclonal antibodies atezolizumab, durvalumab, and avelumab. Current inclusion criteria for antibody monotherapy requires a tumor biopsy and positive staining for the IC [148]. However, despite upregulation and indication across many cancers, the predicted response rate to anti-PD-1/PD-L1 therapy remains 20–30% [149]. Even if patients did initially respond to therapy, many patients developed resistance and relapsed following treatment [150].

In breast cancer, the story is tragically similar. While several subtypes of breast cancer have targetable receptors, patients with triple negative breast cancer (TNBC) are lacking options. Prior to immunotherapeutic options, TNBC patients would receive surgical resection, radiotherapy, chemotherapy or a combination of the three [151]. TNBC patients will experience recurrence at a higher rate than other breast cancers subtypes [152]. Amongst TNBC patients that did not experience recurrence, increased immune system activation against their cancers was discovered [153]. Immunotherapy represents an actionable target for many TNBC patients; PD-L1 specifically is upregulated in up to 20% of TNBC patients [154]. The problem remains that monotherapy with anti-PD-1 or anti-PD-L1 antibodies is not effective on its own. When combined with chemotherapy however, patients receiving anti-PD-1 antibodies showed improved outcomes in the KEYNOTE-355 trial [155]. In the IMPassion131 trial, anti-PD-L1 antibody combined with chemotherapy did not improve survival rates compared to chemotherapy alone [156]. The benefits of targeting of the PD-1/PD-L1 axis remain unclear, some treatments prove to be efficacious while others much less so. In some cases, targeting of this axis has been found to upregulate alternate immune checkpoints as a form of resistance [157]. Improved immunohistochemical methods for the detection of PD-L1 might help in selecting patients with a better expression of PD-L1 in the tumor.

Monoclonal antibodies targeting the PD-1/PD-L1 axis is a well-documented immunotherapy approach that functions after the targets are expressed on the cell surface. However, recent studies have aimed at intracellular regulation of this axis and manipulation of the TME. New developments in targeted proteolysis of oncogenic proteins via Proteolysis Targeting Chimeras (PROTACS) have shown promise in murine breast cancer models [158]. These methods could be employed against the PD-1/PD-L1 axis. Intracellular regulation was directly achieved via metformin administration, which activated AMPK, in turn facilitating endoplasmic reticulum (ER) associated homeostasis pathways such as autophagic degradation of PD-L1 [159,160]. Using chitosan biguanide (Bi-Ch) to target the mitochondria in tumor cells, researchers decreased the available ATP for the multi-drug resistance protein (MDR-1) [161]. Further refining this treatment to a metformin (Ch-met) modified chitosan inhibited MDR-1, while simultaneously decreasing PD-L1 expression through AMP-activated protein kinase (AMPK) activation [162]. Ch-met also potentiated other chemotherapeutic agents such as platinum analogs [162].

## 7. Overcoming the Resistance to Immunotherapy

Monoclonal antibodies targeting ICs have shown very promising outcomes in the field of oncology. Presently, there are multiple FDA-approved immune checkpoint inhibitors (ICIs) being used in the clinic (Table 5). They are divided into four groups: anti PD-1, anti-PD-L1, anti-CTLA-4 and anti-LAG-3 with more ICIs in different phase trials as shown in previous tables.

Despite the impressive outcomes and survival benefits with ICI therapy in various cancer types [163,164,165,166,167], the response rate to ICIs shows a dichotomy in which some patients show a strong response and significant benefit from therapy, while others do not. Moreover, with longer follow-up periods, relapse events are starting to be more common amongst patients. All this suggests at least two types of ICI resistance: primary resistance (or innate), and secondary (or acquired), with some groups call a third type termed “adaptive resistance” (intermediate phenotype characterized by limited anti-tumor response with immunosuppression) [168]. The likelihood of developing primary or secondary resistance increases significantly in patients undergoing ICI monotherapy [169]. Several mechanisms of ICI resistance have been proposed (Figure 3). Cellular pathways of resistance include the impairment of antigen-stimulated T lymphocyte responses, T lymphocyte exhaustion, and cellular expression of alternative ICs. Signaling pathway resistance may stem from disrupted IFN-γ signaling in tumor cells, or the dysregulation of oncogenic signaling pathways (WNT–β-catenin, CDK4/6, MAPK cascades). The emergence of resistance to ICIs is characterized by one or more of the following phenomena: insufficient anti-tumor T lymphocyte generation, inadequate anti-tumor T lymphocyte effector function, or impaired formation of T lymphocyte memory [170]. Tumor heterogeneity also plays a role in resistance against ICI therapy. Different patients with the same type of tumor can display different resistance mechanisms, and within a single patient, multiple resistance mechanisms can co-exist.

### 7.1. Expression of Alternate ICs

Resistance to one ICI is often rooted in the function of other ICs. For example, targeting PD-1 increases the expression of other ICs, such as CTLA-4, TIM-3 and LAG-3 [169,171]. It is important to understand the common transcription and translation factors that regulate the expression of different ICs at a molecular level. TCR/CD28 signaling is regarded as one of the main triggers for IC expression. Upon the activation of T lymphocytes via TCR signaling, the p38MAPK/JNK pathway is activated, which further activates a c-Jun/ATF2 complex [172]. This complex recruits the transcription factor Yin-Yang (YY1). YY1 binds to the to the promoter region of the genes encoding PD-1, LAG-3, and TIM-3 and increases their transcription [172]. Other regulators of the expression of ICs are cytokine and interferon receptors. In TNBC, tumor cells induce the expression of IL-27, which through STAT1/STAT3 signaling activates the transcription factors c-MAF, NFIL3 and BLIMP1, promotes chromatin remodeling at the *TIM-3* locus. This remodeling leads to the increased expression of TIM-3 [173,174]. Furthermore, IL-27 expression is also linked to enhanced activity of PD-1, TIGIT and LAG-3 within the TME via a STAT1/STAT3 signaling cascade [174,175]. Yet another factor that can regulate the expression of ICs is one of the receptors of vascular endothelial growth factor A (VEGF-A) called VEGF receptor 2 (VEGFR2). To induce angiogenesis, tumor cells produce abundant amounts of VEGF-A in the TME. Tumor-infiltrating mature CD8^+^ T lymphocytes in turn express VEGFR2, which interacts with the VEFG-A produced [176]. This interaction sends downstream signals through a PLCγ-calcineurin-NFAT pathway enhancing the production of ICs by activating the genes such as 2B4, CD160, KLRG1, and BTLA, ultimately increasing the expression of PD-1, TIM-3, and LAG-3 [176]. TIM-3 was shown to be upregulated following anti-PD-1 therapy in immunocompetent murine models, and patients who developed adaptive resistance to anti-PD-1 therapy displayed a similar trend with upregulation of TIM-3 [157]. This suggests that sequential therapy or combination therapy with more than one classes of ICIs could alleviate the emergence of resistance after immunotherapy. In line with this, several trials have demonstrated how combination therapy yields superior results compared to monotherapy [177,178] or current treatment guidelines [179].

It is believed that DNA methylation and histone modification also plays a key role in regulating the expression of ICs. In breast cancer, demethylation enzymes TET2 and TET3 were upregulated whereas methylation enzymes such as DNMT3a and DNMT3b were found to be downregulated. Downregulation of methylation enzymes was found to have major impact on the hypomethylation of the promoter regions of *pdcd1* and *tim*-3, thus increasing their transcription. However, no changes in the methylation status were significant for the genes encoding and regulating TIGIT and LAG-3, suggesting that they might follow different regulation patterns [51,180].

### 7.2. Impaired Antigen-Stimulated T Lymphocyte Response

Factors hindering antigen presentation to T lymphocytes, such as downregulation of MHC expression or insufficient generation of antigens, often results in unsatisfactory responses to ICI therapy. A tumor mutational burden (TMB) is the total number of somatic mutations per megabase of genomic sequence [181,182], and it has been proposed as a tool for predicting therapeutic response to ICIs [183,184]. These mutations result in novel protein products called neoantigens, which are then presented to T lymphocytes and generate an immune response. Tumors with a low TMB usually show a limited response or complete resistance to ICIs, most likely due to insufficient neoantigen formation/presentation [185,186,187,188], Conversely, high response rates have been reported in tumors with a high TMB [189]. Further in line, a dramatic robust response was observed in mismatch repair deficiency, in locally advanced rectal cancer showing a high TMB [190]. These antigens are presented to T lymphocytes mainly through major histocompatibility complex (MHC) class I, and a disruption in the antigen processing machinery can lead to decreased T lymphocyte responses and evasion of immune surveillance and cytolytic activity [191]. This disruption of the antigen presentation machinery can be due to a decreased MHC expression, an alteration in the MHC structure, or mutation in the transport of the antigen to the surface [192,193]. Furthermore, alterations of β2-microglobulin (β2M), a structural component of MHC class I, is present in tumors displaying resistance to PD-1 blockade [194], PD-1/CTLA-4 blockade [195], and PD-1/PD-L1 blockade [196].

### 7.3. Disrupted IFN-γ Signaling in Tumor Cells

When tumor-reactive T lymphocytes mount an adequate response against tumor neoantigens, IFN-γ is produced in the TME which activates JAK/STAT signaling, which in turn activates the transcription factor interferon regulatory factor 1 (IRF1), culminating in the transcription of *cd274* (whose protein product is PD-L1). PD-L1 on the surface of tumor cells decreases T lymphocyte anti-tumor activity upon binding PD-1, creating a negative feedback loop that ultimately results in downregulating antitumor responses. PD-1/PD-L1 blockade breaks this cycle and restores antitumor activity. However, when IFN-γ signaling is disrupted, a PD-L1 adaptive response is absent, rendering the PD-1/PD-L1 blockade ineffective [22]. The IFN-γ signaling cascade can be affected by mutations of proteins that form part of this pathway, such as Jak1, Jak2, and Stat1, among others [197,198]. Mutations or loss of IFN-γ genes was identified as the reason for resistance to anti-CTLA-4 therapy [199,200].

### 7.4. T Lymphocyte Exhaustion

Upon encountering an antigen, T lymphocytes mount an adaptive immune response specifically aimed to clear the foreign immunogen. After clearance, a small subset of T lymphocytes differentiates into memory cells with the task of clearing the foreign antigen in the event of being encountered again. However, cancer cells can evade immune clearance and persist in the presence of effector T lymphocytes even after tumor antigens have been recognized. Continued presence of tumor antigens can lead those T lymphocytes to lose their effector function and become exhausted, rendering them ineffective for tumor control [201,202,203]. This dysfunctional state is distinct from functional effector and memory T lymphocytes in both transcriptional terms and expression of ICs [204]. Proliferative capacity, lack of cytotoxic and cytolytic activity, and decreased IFN-γ upon TCR binding are hallmark of exhausted T lymphocytes, which end up being eliminated [205,206]. Interestingly, exhausted T lymphocytes express high levels of PD-1 [136,207,208,209] and other ICs as well, such as TIM-3, LAG-3, and the IC common transcription factor BLIMP1 [209,210,211]. Although anti-PD-1/PD-L1 therapy aims to invigorate and rescue exhausted cells, severe, irreversible exhaustion characterized by an overexpression of ICs, especially PD-1, might be a cause of refractoriness to ICI therapy [201,212,213,214,215]. Conversely, dietary regulation of IFN-γ has been employed as a strategy to combat T lymphocyte exhaustion [216]. By administering a ketogenic diet, researchers were able to down regulate PD-L1 levels and promote IFN-γ expression in the TME, which in turn led to improved efficacy of ICI therapy [216].

### 7.5. Dysregulation of Oncogenic Signaling Pathways

Tumor signaling pathways and their role in shaping the TME have been reviewed extensively [217,218,219]. Evidence suggesting a role in ICI resistance has been described for at least three oncogenic pathways: MAPK, Cyclin-dependent kinases 4 and 6 (CDK4/6) and WNT–β-catenin cascades.

Expression of immunosuppressive cytokines IL-6 and IL-10 is increased by the MAPK signaling pathway, resulting in immune evasion [220]. Consequently, cancers with overactivation of the MAPK pathway may show a decreased response to ICIs due to poor T lymphocyte engagement. The fact that this occurs secondary to an already subpar T lymphocyte recruitment and function further explains resistance. In addition, inhibition of the MAPK pathway using EGFR and MEK inhibitors prevented expression of PD-L1 at the transcriptional and translational levels in non-small cell lung cancer (NSCLC) [221]. Further corroborating the link between PD-L1 and MAPK, MAPK signaling was shown to regulate PD-L1 expression in lung cancer [222]. As previously described, the MAPK pathway leads to the recruitment of transcription factor YY1, which regulates expression of PD-1, LAG-3 and TIM-3 [174]. Anti-PD-1 therapy combined with MEK inhibition produced lasting tumor regression in CT26-innoculated BALB/c mice [223] and the triple therapy with trametinib (MEK inhibitor), dabrafenib (BRAF inhibitor) and anti-PD-1 showed a strong anti-tumor effect [224]. Collectively, MAPK can lead to ICI resistance through many discovered pathways: through decreasing T lymphocyte activity via IL-6 and IL-10; by increasing expression of PD-L1 on tumor cells; and by promoting the co-expression of several ICs in T lymphocytes. Overcoming this form of resistance may lay in a parallel signal transduction pathway. The interplay between the MAPK pathway and the AMP protein kinase (AMPK) pathway within the TME has been extensively studied [225]. The overactivation of the MAPK pathway leads to the suppression of the AMPK pathway [225]. Recently, studies have shown that pharmacological activation of the AMPK pathway leads to a downregulation of the PD-L1 within the TME [226,227].

CDKs are a family of critical regulatory enzymes that integrate intra- and extracellular cues to control cell proliferation by regulating the progression through the different phases of the cell cycle and modulating transcription [228]. Given their importance, mutation and dysregulation of CDKs are often implicated in diseases such as cancer. CDK4/6 inhibitors now constitute a widely used class of anticancer agents with 3 FDA-approved members (as of October 2022), abemaciclib, albociclib and ribociclib. The role of CDK4/6 inhibitors in anti-tumor immunity has been corroborated by several studies [229,230,231,232]. Analysis of single-cell RNA sequence from melanoma patients identified an ICI resistance program that was controlled by CDK4/6 [232]. After treatment with abemaciclib, this program was repressed, and the melanoma cells were re-sensitized to anti-PD-1 therapy in mouse models. In another study, CDK4/6 inhibitors in combination with anti-PD-L1 therapy was more effective than either individual agent in murine breast cancer models [231]. In general, CDK4/6 inhibitors enhance the anti-tumor immune response by promoting antigen processing and presentation, stimulating DC maturation, and reducing the population of immunosuppressive T_regs_ [188]. A growing body of evidence suggests that dysregulation of CDK4/6 functions may lead to ICI resistance.

The canonical WNT–β-catenin pathway has been implicated in hindering anti-tumor immune responses and fostering an immunologically “cold” TME by inducing the production of immunosuppressive IL-10 by tumor cells. This impairs the maturation of DCs and pushes immature DCs to differentiate into regulatory DCs. This regulation promotes immune tolerance and produces more IL-10 [233,234,235], creating a feedback loop of immunosuppression [236]. In addition, the WNT–β-catenin pathway has been implicated in decreasing T lymphocyte infiltration and promoting resistance to ICIs. In melanoma, this pathway interferes with the recruitment of DCs and decreases the expression of CCL4 [237,238]. The priming of anti-tumor CD8^+^ T lymphocytes is dependent on this subpopulation of DCs [239]. Melanoma cells can release the soluble WNT agonist WNT5A, that in turn activating the WNT–β-catenin pathway in DCs. This upregulates the activity of the immunomodulatory enzyme indoleamine 2,3-dioxygenase-1 (IDO1). IDO1 converts the amino acid tryptophan into kynurenine, and this increased kynurenine promotes the differentiation of T_regs_ [240]. IDO1 expression is also controlled by IFN-γ signaling [241,242]. The serine/threonine p21-activated kinase 4 (PAK4) is a mediator of the WNT–β-catenin pathway with pro-oncogenic functions such as cell growth, invasiveness, survival, and apoptosis resistance. Elevated levels of PAK4 have been found in several types of cancer [243]. Resistance to PD-1 blockade was overcome after genetic deletion of PAK4, and the combination of anti-PD-1 with PAK4 inhibition improved anti-tumor response versus PD-1 blockade alone [244].

## 8. Conclusions

Immunotherapy is better for patients in terms of the number and the extent of the adverse events following therapy. The classical ICs and the activation of the alternate ICs must be borne in mind when employing immunotherapy. Precision medicine following the evaluation of ICs through biopsy along with a low dose cytotoxic or targeted therapy might pave way for a better clinical outcome in treating therapy-naïve or immuno-refractory cancers.

## Figures and Tables

**Figure 1 cancers-15-02774-f001:**
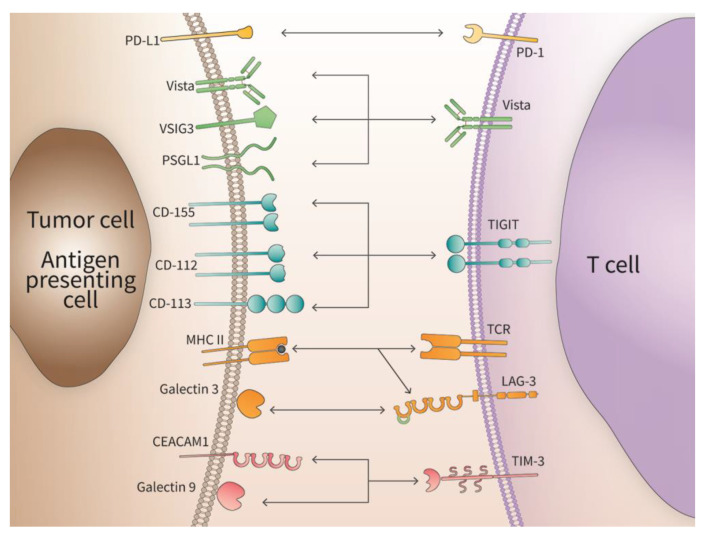
Interaction of immune checkpoint ligands and their cognate receptors. The specific interaction of immune check point receptors with their respective ligands is depicted. Some receptors have a plurality of ligands indicating a context-dependent function.

**Figure 2 cancers-15-02774-f002:**
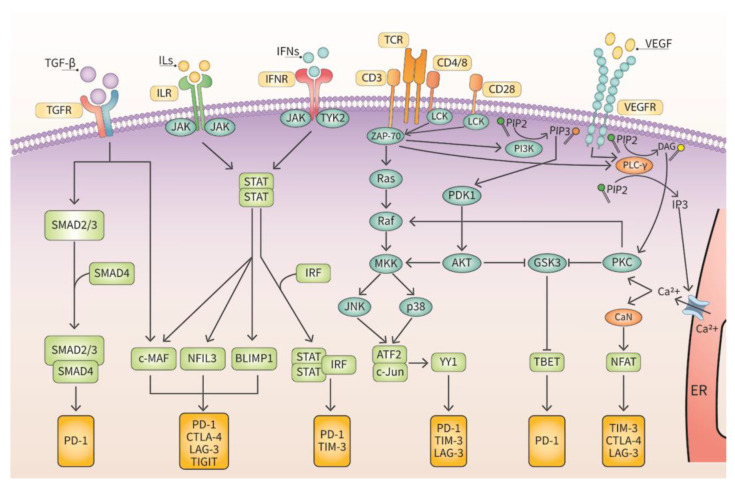
Cell signaling cascades induce transcription of ICs. Several cell signaling pathways either directly activate the transcription of ICs or indirectly through epigenetic pathways.

**Figure 3 cancers-15-02774-f003:**
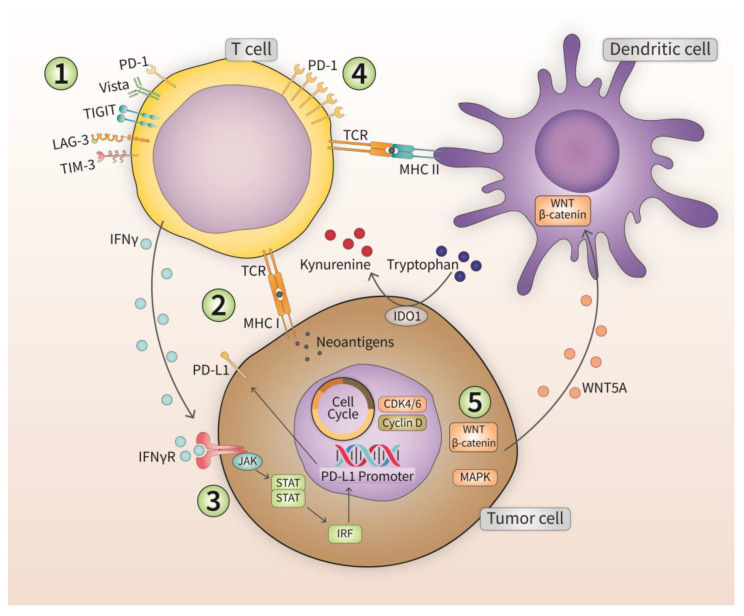
Different mechanisms employed by tumor cells to evade immunotherapy. Five different mechanisms are discussed in the section outlining the development of resistance in immuno-refractory tumors.

**Table 1 cancers-15-02774-t001:** The clinical trials of anti-VISTA therapy drugs.

Drug Name	ClinicalTrials.gov Identifier (NCT)	Phase	Combined with	Targets
CI-8993 [48]	NCT04475523	Phase 1	-	Solid tumors
CA-170 [49]	NCT02812875	Phase 1 completed	-	Advance solid tumors and lymphomas
HMBD-002 [50]	NCT05082610	Phase 1	Pembrolizumab (anti-PD-1)	Advance solid malignancy

**Table 2 cancers-15-02774-t002:** Anti-TIGIT antibodies targeting breast cancers.

Drug Name	ClinicalTrials.gov Identifier (NCT)	Phase	Combined with	Targets
JS006 [71]	NCT05061628	Phase 1	Toripalimab	Advanced tumors
IBI939 [72]	NCT04672369	Phase 1	Sintilimab	Advanced malignancies
COM902 [73]	NCT04354246	Phase 1	-	Advanced malignancies
M6223 [74]	NCT04457778	Phase 1	Bintrafusp alfa	Metastatic solid tumors
BAT6021 [75]	NCT05073484	Phase 1	BAT1308 (anti-PD1)	Advanced solid tumors
Domvanalimab [76]	NCT05502237	Phase 1	Zimberelimab (anti-PD1)	Solid tumors
COM701 [77]	NCT04570839	Phase 1	Nivolumab (anti-PD1)	Advanced cancers
NTX-1088 [78]	NCT05378425	Phase 1	Pembrolizumab (anti-PD1)	Cancer
OMP-313M32 [79]	NCT03119428	Phase 1	Nivolumab (anti-PD1)	Metastatic cancer
BMS-986207 [80]	NCT02913313	Phase 2	Nivolumab Ipilimumab	Broad solid tumors
Tiragolumab [81]	NCT04294810	Phase 1	Atezolizumab Nab-paclitaxel Carboplatin	TNBC

**Table 3 cancers-15-02774-t003:** Anti-LAG-3 antibodies targeting breast cancers.

Drug Name	ClinicalTrials.gov Identifier (NCT)	Phase	Combined with	Targets
BI754111 [97]	NCT03156114	Phase 1	BI754091 (anti-PD1)	Carcinoma
LAG525 [98]	NCT03365791	Phase 2	PDR001 (anti-PD1)	TNBC

**Table 4 cancers-15-02774-t004:** Anti-TIM3 agents in cancer clinical trials.

Drug Name	ClinicalTrials.gov Identifier (NCT)	Phase	Combined with	Targets
Sym023 [123]	NCT03489343	Phase 1	-	Metastatic cancers
MBG453 [124]	NCT04266301	Phase 2	PDR001 (anti-PD1) Decitabine	Advance malignancies
LY3321367 [125]	NCT03099109	Phase 1	-	Solid tumor
RO7121661 [126]	NCT04785820	Phase 1	-	Solid tumor
LB1410 [127]	NCT05357651	Phase 1	-	Solid tumor
TSR-022 [128]	NCT03680508.	Phase 1	TSR-042 (anti-PD1)	Advance solid tumor
BGBA425 [129]	NCT03744468	Phase 1	Tislelizumab (anti-PD1)	Metastatic tumors
INCAGN02390 [130]	NCT03652077	Phase 1	-	Solid tumors
BMS-986258 [131]	NCT03446040	Phase 1	Nivolumab	Advanced cancer

**Table 5 cancers-15-02774-t005:** FDA-approved ICIs.

Drug	Class	First FDA Approval Date
Nivolumab	Anti-PD-1	22 December 2014
Pembrolizumab		4 September 2014
Cemiplimab		28 September 2018
Atezolizumab	Anti-PD-L1	18 May 2016
Avelumab		23 March 2017
Durvalumab		1 May 2017
Ipilimumab	Anti-CTLA-4	28 March 2011
Tremelimumab		24 October 2022
Relatlimab	Anti-LAG-3	18 March 2022

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
