# Peer review of "Tackling of Immunorefractory Tumors by Targeting Alternative Immune Checkpoints"

_cancers, 2023, doi:10.3390/cancers15102774_

Round 1

Reviewer 1 Report

Comments

·       The authors have attempted to compile the vast literature of multiple immune checkpoint proteins other than PD-1 and CTLA4 (which are the common ones) which is good.

·       The review article gives an overall idea of these proteins in multiple tumors and information with focus on breast cancer in particular is kind of less when compared to the title of the article. So it’s better to rephrase the title and mention about the inclination towards breast cancer in the content.

·       Abstract – The sentence “The ICs are…. unabated” is unclear and has to be rewritten. The abstract could have a sentence on what is the uniqueness of this particular review article and highlight the importance of the content of this paper to give a better idea to the reader.

·       Introduction – Page1, lines 22-23 are unclear.

1.     Page2, line 48 – the sentence “IFN-γ …blood flow” must be rephrased as it does not convey the exact meaning of what the authors want to convey.

2.     Page2, lines 54-55 – Authors mention that “cytotoxic CD8+ T lymphocytes directly contribute to tumor cytotoxicity through secretions of perforin and granzyme cytokines”. Bothe CD4+ and CD8+ cytotoxic T cells secrete perforin and granzyme B. So the sentence must be edited accordingly.

3.     Page 2, line 63 – should be “allows the regulation” instead of “allows for regulation”. In the same line, the sentence “Immune cells are not always in control” is unclear.

4.     Page 4, line 111 – the sentence must be “…Th1 T lymphocytes and was described as a negative regulator of immune cells”. Similarly line 116, sentence must be “Eventually, the finding that PD-L1 expressing normal somatic cells”

5.     Page 5, line 164 – the sentence must be “One mechanism by which VISTA promotes”. Lines 173-176, the sentence is too long. Lines 176-177 is a repeat of information.

6.     Page 6, lines 178-179, TME is repeated. Sentence has to be edited. Line 180 – is again a repeat of information. Lines 193-194 – the sentence must be “MDSCs …..to either macrophage or neutrophil lineage”. Lines 196-199 – Sentence too long. Lines 206-207 should be rephrased to convey the correct meaning.

7.     Page 8, line 281 - foxp3 should be corrected as “Foxp3”

8.     Page9, lines 302-305 is not necessary. Direct description of LAG3 would be more appropriate.

9.     Page 10 line 336 – “binding site of LAG-3 for CD4 on CD4 T lymphocytes” is unclear. Line 337 – the sentence should be “Another ligand of LAG-3 is Gal-3”. Line 341 – the statement “tumor cells in TIME” is unclear. Lines 352-353 – Sentence is incomplete. Line 373 – “expressed in non-T lymphocyte leukocytes” is unclear. In the same line it should be mast cells and not “masT lymphocytes”.

10.  Page 11, lines 409-410 – Sentence is unclear.

·       Overcoming the resistance to immunotherapy - Page 16 lines 558-560 – The information is repeated. Line 564 – the phrase “or standard of care” needs to be explained better.

1.     Page 17, line 623 – Sentence should be “their role shaping in TME”. Lines 628-633 and 645-648 – Sentence is unclear and too long. Line 645, the sentence must be corrected as “…single agent in mouse breast cancer models”

2.     Page 18, lines 661-664 – Sentence is too long.

·       Including the information on the mode of action for the drugs mentioned in the tables could improve the understanding and relevance of these drugs.

·       Throughout the paper, uniform abbreviations are needed to be followed. For example, tumor microenvironment is mentioned as “TIME” in few places and “TME” in few places; similarly interferon-g is mentioned as “IFN” and “INF” Also MHC abbreviation has been mentioned already in the article, but has been abbreviated again. All the words used as abbreviations and mentioned in the list can be followed uniformly throughout the draft.

·       Throughout the paper, English language has to be checked to have uniform tense in the draft. Sudden change from present to past perfect is there in multiple places and also the use of “The” in appropriate places is suggested.

Author Response

Reviewer1 - Comments · The authors have attempted to compile the vast literature of multiple immune checkpoint proteins other than PD-1 and CTLA4 (which are the common ones) which is good. Thank you very much! · The review article gives an overall idea of these proteins in multiple tumors and information with focus on breast cancer in particular is kind of less when compared to the title of the article. So it’s better to rephrase the title and mention about the inclination towards breast cancer in the content. We have rephrased the title accordingly, thanks! · Abstract – The sentence “The ICs are…. unabated” is unclear and has to be rewritten. The abstract could have a sentence on what is the uniqueness of this particular review article and highlight the importance of the content of this paper to give a better idea to the reader. Good point. We have revised the abstract to reflect the uniqueness of this review, thanks! · Introduction – Page1, lines 22-23 are unclear. These have been modified to make it clear now. 1. Page2, line 48 – the sentence “IFN-γ …blood flow” must be rephrased as it does not convey the exact meaning of what the authors want to convey. Yes, we have rephrased the sentence now, thanks! 2. Page2, lines 54-55 – Authors mention that “cytotoxic CD8+ T lymphocytes directly contribute to tumor cytotoxicity through secretions of perforin and granzyme cytokines”. Bothe CD4+ and CD8+ cytotoxic T cells secrete perforin and granzyme B. So the sentence must be edited accordingly. Thanks for pointing this out! We have edited it accordingly now! 3. Page 2, line 63 – should be “allows the regulation” instead of “allows for regulation”. In the same line, the sentence “Immune cells are not always in control” is unclear. We edited it to make it clear now, thanks! 4. Page 4, line 111 – the sentence must be “…Th1 T lymphocytes and was described as a negative regulator of immune cells”. Similarly line 116, sentence must be “Eventually, the finding that PD-L1 expressing normal somatic cells” Yes, we have changed this now, thanks! 5. Page 5, line 164 – the sentence must be “One mechanism by which VISTA promotes”. Lines 173-176, the sentence is too long. Lines 176-177 is a repeat of information.
We have now cut the long sentence into short sentences. Repeat of information is deleted now, thanks!
6. Page 6, lines 178-179, TME is repeated. Sentence has to be edited. Line 180 – is again a repeat of information. Lines 193-194 – the sentence must be “MDSCs …..to either macrophage or neutrophil lineage”. Lines 196-199 – Sentence too long. Lines 206-207 should be rephrased to convey the correct meaning. Yes, we have rephrased this now, thanks! 7. Page 8, line 281 - foxp3 should be corrected as “Foxp3” This was corrected now, thanks!
8. Page9, lines 302-305 is not necessary. Direct description of LAG3 would be more appropriate. We deleted the sentence and directly described LAG3 now, thanks! 9. Page 10 line 336 – “binding site of LAG-3 for CD4 on CD4 T lymphocytes” is unclear. Line 337 – the sentence should be “Another ligand of LAG-3 is Gal-3”. Line 341 – the statement “tumor cells in TIME” is unclear. Lines 352-353 – Sentence is incomplete. Line 373 – “expressed in non-T lymphocyte leukocytes” is unclear. In the same line it should be mast cells and not “masT lymphocytes”. All of these suggestions were good. We have changed them now, thanks!
10. Page 11, lines 409-410 – Sentence is unclear. We made it clear now, thanks! · Overcoming the resistance to immunotherapy - Page 16 lines 558-560 – The information is repeated. Line 564 – the phrase “or standard of care” needs to be explained better. We explained it better now, thanks! 1. Page 17, line 623 – Sentence should be “their role shaping in TME”. Lines 628-633 and 645-648 – Sentence is unclear and too long. Line 645, the sentence must be corrected as “…single agent in mouse breast cancer models”.
We corrected it now, thanks! 2. Page 18, lines 661-664 – Sentence is too long. We made the sentence more apt now, thanks!
· Including the information on the mode of action for the drugs mentioned in the tables could improve the understanding and relevance of these drugs. We have included this information now, thanks! · Throughout the paper, uniform abbreviations are needed to be followed. For example, tumor microenvironment is mentioned as “TIME” in few places and “TME” in few places; similarly interferon-g is mentioned as “IFN” and “INF” Also MHC abbreviation has been mentioned already in the article, but has been abbreviated again. All the words used as abbreviations and mentioned in the list can be followed uniformly throughout the draft. Yes, we made changes to reflect the suggestions by the reviewer, thanks! · Throughout the paper, English language has to be checked to have uniform tense in the draft. Sudden change from present to past perfect is there in multiple places and also the use of “The” in appropriate places is suggested. We edited this article using a native English speaker and grammar was taken care of throughout the manuscript, thanks!

Reviewer 2 Report

      In this research, the authors reviewed the recent development of Tackling of immunorefractory breast tumors by targeting alternative immune checkpoints. Generally, it’s meaningful and interesting review. In my opinion, the current version of this manuscript fits the scope of Cancers and could be accepted after major revision.

My specific comments are in detail listed below:

1.     No references were added to the tables. In my opinion, references should be added to support their opinion.

2.     Some minor mistakes existed in this paper. The authors should carefully check it.

3.     In this review, the novel pathways that could affect PD-L1 expression to disrupt PD-1/PD-L1 axis rather than only PD-L1 antibodies or PD-1 antibodies should be added. Some more relevant references should be added including 10.1016/j.carbpol.2022.119878, 10.1038/s41467-022-32050-4, 10.1016/j.jconrel.2022.11.004,   and 10.1016/j.ijbiomac.2022.10.167.

4.     The quality of the figures may still could be improved.

5.     In Line 593-604, how IFN-γ affect the expression of PD-L1 after different treatments should be reviewed or added. Some more relevant references should be added 10.1016/j.carbpol.2021.118869, 10.1016/j.molcel.2021.03.037, 10.1016/j.cej.2022.140164, and 10.1016/j.apsb.2022.07.023.

6.     Better future perspective should be added, especially the prospects of small molecules targeting different immune pathways including CTLA-4, PD-1, VISTA, LAG-3, TIGIT and TIM-3 for clinical transformation.

Author Response

Reviwer2-peer-review In this research, the authors reviewed the recent development of Tackling of immunorefractory breast tumors by targeting alternative immune checkpoints. Generally, it’s meaningful and interesting review. In my opinion, the current version of this manuscript fits the scope of Cancers and could be accepted after major revision. My specific comments are in detail listed below: 1. No references were added to the tables. In my opinion, references should be added to support their opinion. Thanks for pointing this out! We now added references for each item/drug/antibody in all of the tables! Excellent suggestion! 2. Some minor mistakes existed in this paper. The authors should carefully check it. We carefully reviewed the manuscript, and a native English speaker also corrected the grammar. Thanks! 3. In this review, the novel pathways that could affect PD-L1 expression to disrupt PD-1/PD-L1 axis rather than only PD-L1 antibodies or PD-1 antibodies should be added. Some more relevant references should be added including 10.1016/j.carbpol.2022.119878, 10.1038/s41467-022-32050-4, 10.1016/j.jconrel.2022.11.004, and 10.1016/j.ijbiomac.2022.10.167. We have added the suggested references. We added the information requested, thanks! 4. The quality of the figures may still could be improved. Sorry, we had very high quality figures, but the manuscript allows only certain size! Our figures were drawn in Adobe Illustrator which is of great quality each figure running above 100MB in size. I wish the journal allows us to use a high resolution figure! 5. In Line 593-604, how IFN-γ affect the expression of PD-L1 after different treatments should be reviewed or added. Some more relevant references should be added 10.1016/j.carbpol.2021.118869, 10.1016/j.molcel.2021.03.037, 10.1016/j.cej.2022.140164, and 10.1016/j.apsb.2022.07.023. Thanks for pointing this out! We have added the suggested references. We added the information requested and it is highlighted now, thanks! 6. Better future perspective should be added, especially the prospects of small molecules targeting different immune pathways including CTLA-4, PD-1, VISTA, LAG-3, TIGIT and TIM-3 for clinical transformation. We have added the information at appropriate places (all highlighted), thanks

Round 2

Reviewer 2 Report

The current version of this paper could be accepted.